# Efficacy and Toxicity of Classical Immunosuppressants, Retinoids and Biologics in Hidradenitis Suppurativa

**DOI:** 10.3390/jcm11030670

**Published:** 2022-01-27

**Authors:** Kinnor Das, Steven Daveluy, George Kroumpouzos, Komal Agarwal, Indrashis Podder, Katherine Farnbach, Alex G. Ortega-Loayza, Jacek C. Szepietowski, Stephan Grabbe, Mohamad Goldust

**Affiliations:** 1Department of Dermatology Venereology and Leprosy, Silchar Medical College, Silchar 788014, India; daskinnor.das@gmail.com; 2Department of Dermatology, Wayne State University, Detroit, MI 48202, USA; sdaveluy@med.wayne.edu; 3Department of Dermatology, Alpert Medical School of Brown University, Providence, RI 02903, USA; George.Kroumpouzos@gkderm.com; 4Department of Dermatology, Medical School of Jundiaí, São Paulo 13202-550, Brazil; 5GK Dermatology, P.C, Weymouth, MA 02190, USA; 6Department of Dermatology, CNMC, Kolkata 700014, India; agarwall.komal@gmail.com; 7Department of Dermatology, Venereology and Leprosy, College of Medicine and Sagore Dutta Hospital, Kolkata 700058, India; ipodder88@gmail.com; 8Department of Dermatology, Oregon Health and Science University, Portland, OR 97239, USA; farnbach@ohsu.edu (K.F.); ortegalo@ohsu.edu (A.G.O.-L.); 9Department of Dermatology, Venereology and Allergology, Wroclaw Medical University, 50-367 Wroclaw, Poland; 10Department of Dermatology, University Medical Center Mainz, 55131 Mainz, Germany; stephan.grabbe@unimedizin-mainz.de

**Keywords:** hidradenitis suppurativa, immunosuppressants, treatment

## Abstract

Hidradenitis suppurativa (HS) is a chronic inflammatory disorder of the apocrine glands characterized by recurrent episodes. Although several therapies exist, none is completely curative. Several immunosuppressives have been studied with encouraging results and targeted approaches. In this review, we highlight the various immunosuppressives used in this condition along with their salient features to enable physicians to choose the correct therapy for their patients. The search of the peer-reviewed literature included clinical trials, scientific reviews, case series, case reports, and guidelines. The literature was identified from electronic databases (MEDLINE and PubMed) through November 2021; additional articles were included from the references of the identified articles.

## 1. Introduction

Hidradenitis suppurativa (HS) is a debilitating and relapsing inflammatory disease of the apocrine sweat glands. The clinical hallmark is tender, deep-seated, inflamed nodules, which heal with scarring. Commonly affected sites include the axillae, groin, and perineal regions, and mammary, inframammary, and intermammary areas [1]. Most cases occur post-puberty with a predilection for axillary and inguinal folds in females, compared to the perineum and buttocks in males. Genetic predisposition is important as a positive family history has been reported in almost 40% of patients, the most common mode of inheritance being autosomal dominant [2].

Loss-of-function mutations of the gamma secretase gene is considered an important contributory factor for this disease [3]. Other important predisposing factors include local mechanical irritation and shear forces; lifestyle factors (such as obesity, smoking, and tight clothing); certain drugs including sirolimus and lithium [2]. Recently, several authors have highlighted bacterial infections as an etiologic factor for HS. Several bacterial groups causing skin and soft tissue have been implicated such as *Staphylococcus aureus*, *Staphylococcus lugdunensis* [4], other coagulase negative staphylococci (CONS), and mixed anaerobic bacteria, e.g., *Corynebacterium* sp. [5,6]. Furthermore, biofilm formation has been reported in HS lesions, which might result in an inadequate antimicrobial response. Commensal bacteria of the skin, e.g., *Staphylococcus epidermidis* may elicit an inflammatory response and initiate or worsen HS in genetically susceptible individuals [5]. A recent systematic review has speculated about the role of cutaneous microbial dysbiosis in HS pathogenesis [7]. HS may be associated with other cutaneous and systemic comorbidities such as acne conglobata, dissecting folliculitis of the scalp, perifolliculitis capitis abscedens et suffodiens, pilonidal sinus (folliculitis occlusion triad/tetrad), Crohn’s disease, seronegative, and seropositive spondyloarthropathy, pyoderma gangrenosum, SAPHO (synovitis, acne, pustulosis, hyperostosis, and osteitis) syndrome, PAPA (Pyogenic Arthritis, Pyoderma gangrenosum, and acne) syndrome, PAPASH (pyogenic arthritis, pyoderma gangrenosum, acne vulgaris, and HS) syndrome, and genodermatoses such as KID (keratosis ichthyosis deafness syndrome), pachyonychia congenital, and steatocystoma multiplex [1,2].

Hurley staging is used to grade disease severity, estimate prognosis, and measure therapeutic response. Hurley stage 1 is defined by the presence of isolated abscess (es) without scarring or sinus tract formation. In Hurley stage 2, recurrent abscess formation is associated with sinus tract formation and scarring. Hurley stage 3 indicates widespread involvement with multiple interconnected sinus tracts or abscesses involving an entire area [8].

As a definitive cure for HS is lacking; management is symptomatic and remains challenging due to the relapsing and remitting nature of the disease. Various therapeutic modalities have been employed but a “gold standard” treatment remains elusive. Recently, several immunosuppressive drugs have been studied because of their targeted mechanisms, with encouraging results [9]. Management is guided by clinical presentation of the disease; oral immunosuppressants are usually indicated in Hurley stages 2 and 3. In refractory cases, surgical de-roofing of the affected area is necessary for symptomatic relief, especially those cases with biofilm formation not responding to medical management. This review highlights the various immunosuppressants studied in HS to date, along with their dosage, adverse effects, and special considerations. The review is focused on immunosuppressant agents and biologics and other non-antibiotic systemic treatments for HS. Antibiotic and surgical therapy is beyond the scope of this review.

## 2. Methods

The review of peer-reviewed literature included clinical trials and scientific reviews. We searched across multiple databases (MEDLINE and PubMed) using the keywords “Hidradenitis suppurativa” AND “Immunosuppressives” OR “Immunomodulators” OR “Treatment” OR “Therapy”. We included only English language articles without any time limitation. The reference list of individual articles was scanned to identify additional articles.

## 3. Results

The medical treatment options for HS are broadly divided into two types: topical and systemic agents. Topical therapies include topical disinfectants (e.g., triclosan), topical antibiotics (e.g., clindamycin), ammonium bituminosulfonate, and topical immunosuppressants (e.g., intralesional steroids). Systemic treatments include antibiotics, biologics, and retinoids. The examples of useful antibiotics in the treatment of HS include tetracyclines, rifampicin/clindamycin. Adalimumab, an anti-TNF, is an example of a biologic agent used in the treatment of HS. Acitretin is a systemic retinoid for the treatment of HS. Considering its teratogenic potential, retinoids should be used with precaution in women of childbearing age. Other systemic treatment options that need more expertise include dapsone, metformin, systemic corticosteroids, and cyclosporine [10].

Table 1 highlights the various immunosuppressives used in HS along with their mechanism of action and other characteristics [9,11,12]. Each of these drugs is briefly discussed below along with the relevant literature.

### 3.1. Azathioprine (AZA)

Azathioprine is both an immunosuppressive and immunomodulatory molecule. The functioning metabolite of azathioprine, 6-thioguaninemonophosphate, is similar to endogenous purine and integrates into cellular DNA, subsequently inhibiting cell replication. Azathioprine also modulates the function of T cells, B cells, and antigen-presenting cells (APCs) [13].

As an anomalous immune response is considered a key factor in the pathogenesis of HS, and the inflammatory lesions are typified by a concentrated inflammatory infiltrate, it is speculated that azathioprine may be effective in HS. A study conducted by Nazary et al. reported no significant benefit with oral azathioprine (50–100 mg, daily) [14]. However, Martínez et al. highlighted azathioprine as an effective maintenance agent in a single HS patient with concomitant Crohn’s disease, where remission was induced by an anti-tumor necrosis factor [15].

While azathioprine may offer some benefit, long-term use may lead to myelosuppression (especially in patients with low thiopurine methyltransferase level), hepatotoxicity, gastric irritation, and hypersensitivity syndromes (urticaria, morbilliform eruption, purpura, erythema multiforme, and angioedema). It may also increase susceptibility to infections such as herpes simplex and human papilloma virus. Azathioprine is a pregnancy category D drug, contraindicated in pregnancy and best avoided during lactation as it is excreted in breastmilk and colostrum. Overall, there is not sufficient evidence for the use of azathioprine in HS.

### 3.2. Cyclosporine (CsA)

Cyclosporine is a calcineurin inhibitor extracted from the fungus *Tolypocladium inflatum gams*. It represses both cell-mediated and antibody-mediated immune function through a reduction in the secretion of interleukin-2 (IL-2) among other pro-inflammatory factors and interference with Langerhans cell function. Chiefly, it interrupts T cell function [16].

CsA is used at a dose of 3–5 mg/kg/day orally. Common adverse effects include renal dysfunction, hypertension, gingival hyperplasia, arthralgia, hyperuricemia, and hyperkalemia. Regular monitoring of renal function, serum electrolytes, and blood pressure is needed during therapy.

Anderson and colleagues reported minor improvement in lesions of HS with cyclosporine, although the claim remains uncorroborated [17]. No randomized controlled trials have been conducted regarding CsA in HS. Furthermore, there are no head-to-head studies of CsA with other drugs used in HS treatment. It is a pregnancy category C drug, best avoided in lactation and in children under 18 years of age. According to European guidelines, CsA is reserved for the treatment of patients who do not respond to standard first-, second- or third-line treatment. Dosage for the treatment of HS ranges from 2 to 6 mg/kg, whereas the duration range is 1.5 to 7 months [18].

### 3.3. Corticosteroids

Corticosteroids exert an anti-inflammatory effect through interruption of the synthesis of prostaglandins, leukotrienes, and cytokines, leading to the inhibition of leucocyte advancement to the sites of inflammation.

The response of HS to oral corticosteroids is variable [18]. Oral corticosteroids are often used for the control of acute inflammatory flare ups in combination with other topicals and/or immunosuppressants, and result in improvement [11]. The suggested dose of prednisolone is 0.5–0.7 mg/kg/day [10].

However, in some cases the withdrawal of corticosteroids may lead to relapse. Hence, corticosteroids administered in high doses are not much use in long-term management of HS. In a small retrospective study, Wong et al. reported the efficacy of low-dose prednisone as a long-term adjunct therapy along with other medical therapies [19]. In this small study, 13 patients with recalcitrant disease (mean duration of disease 10.8 years) were treated with low dose prednisone. Out of them, five patients showed remission with addition of low dose prednisone after 4 to 12 weeks. Three of them had remission for up to 6 months after stopping prednisolone. Another six patients had partial response with persistence of only minor lesions [19]. These results suggest a possible synergistic effect of low dose corticosteroid with other therapies.

Oral corticosteroids have an array of possible side effects including gastritis, headaches, weight gain, Cushing syndrome, cataracts, dyslipidemia, hypertension, diabetes mellitus, depression, and increased susceptibility to infections, requiring regular monitoring of vitals and laboratory studies. It is a pregnancy category C drug. European guidelines recommend oral prednisolone 0.5–0.7 mg/kg for short-term use for the management of acute flare ups. It should be tapered over the following week [18]. Thus, corticosteroid dosing in HS relies on clinicians’ assessment and low doses may be useful [20].

### 3.4. Dapsone (DDS)

Dapsone is chemically diaminodiphenyl sulphone (DDS) used in HS for its anti-bacterial and anti-inflammatory properties. The anti-bacterial mechanism of dapsone stems from its blockage of bacterial dihydrofolic acid through competitive inhibition of the enzyme dihydropteroate synthase, thus hindering nucleic acid synthesis. The anti-inflammatory mechanism of dapsone is due to the inhibition of myeloperoxidase in neutrophils [21,22]. A study by Yazdanyar et al. reported only 9 of 24 HS patients improved with DDS, and suggested the benefits of DDS are likely to be limited to milder cases of HS [23]. In contrast, Kaur and Lewis found improvement with DDS in all five HS patients in their cohort [24].

Dapsone been used at a dose of 25–150 mg/day orally. Common side effects include nausea, headache, insomnia, leukopenia, agranulocytosis, hemolytic anaemia, peripheral neuritis, hepatitis, and drug hypersensitivity syndrome. It is a pregnancy category C drug and is contraindicated in individuals with G6PD enzyme deficiency.

According to European guidelines, it is reserved for the treatment of patients with a mild–moderate condition after failure of first- and second-line standard therapies [18].

According to a Swiss expert panel, dapsone is recommended for the treatment of follicular occlusion in moderate to severe cases. Similarly, in chronic moderate to severe cases, dapsone is recommended [9].

### 3.5. Methotrexate (MTX)

Methotrexate is an antimetabolite and anti-folate drug. Acting on the S phase of the cell cycle, Methotrexate interferes with the synthesis and repair of DNA through the inhibition of dihydrofolic acid reductase, thus interrupting cellular replication [25]. A single center retrospective study involving 15 patients received MTX at the mean dose of 10 mg/week (range 7.5–20 mg/week) with a mean cumulative dose of 520.1 mg and mean duration of treatment of 11.7 months. Out of these 15 patients, 8 received it with biologics, whereas 7 received it without biologics. Oral antibiotics (*n* = 9) was the most common therapy in patients receiving MTX. Patients using it as a primary therapy showed a non-significant decrease in patient global assessment, inflammatory nodule count, and abscess count. Patients on concomitant biologic therapy did not show a change in these parameters. This study concluded that MTX may be a useful option in older patients with lower body mass indices, but had no benefit in those receiving concurrent biologic therapy [26].

In a report of three patients treated with MTX by Jemec, two patients were treated with 12.5 mg and one patient with 15 mg MTX for 4 months and 6 months, respectively. Neither the primary lesions nor the frequency of subsequent flare ups improved with MTX, as evaluated by both patient and dermatologist [27].

Some studies have shown MTX to be effective in HS associated with other diseases, such as the successful treatment of SAPHO syndrome and HS with infliximab and methotrexate [28]. Common adverse effects include hepatotoxicity and myelosuppression. As a pregnancy category X drug, women of child bearing age on methotrexate should use proper contraception, and men on methotrexate should be counselled regarding possible reversible oligospermia.

It is not recommended by the standard guidelines for the management of HS.

### 3.6. Retinoids

Isotretinoin is also known as 13-cis-retinoic acid. While the precise mechanism of action of isotretinoin is undetermined, numerous studies have revealed that isotretinoin promotes apoptosis of different cell types in the body and down-regulates the enzyme telomerase reverse transcriptase [29,30,31].

Acitretin is a second-generation retinoid with the capacity to activate but not bind to multiple retinoic acid receptors (RARs), inhibiting the inflammatory response and exerting anti-proliferative effects. It stabilizes keratinocyte differentiation and blocks the expression of cytokines such as interleukin-6 (IL-6), migration inhibitory factor-related protein-8 (MRP-8), and interferon-gamma. Acitretin is considered to be superior to isotretinoin for HS management. HS accompanied by nodulocystic acne responds well to acitretin [32,33]. Hogan found acitretin effective in 5 HS patients [33]. In 12 patients with HS treated with acitretin (mean dose 0.59 mg/kg/d) over an average period of 10.8 months, Boer and Nazary reported that 9 of 12 patients showed significant improvement, while the remaining three patients had mild to moderate recovery. Long-term efficacy of acitretin was evaluated in 12 patients with severe and recalcitrant HS. These patients received acitretin for 9–12 months and were followed up to 4 years. All of these patients achieved remission and had a significant reduction in pain [34].

Alitretinoin is a pan-retinoic acid agonist. Anti-inflammatory effects of alitretinoin are exerted by the suppression of chemokine receptor expression and inhibition of inflammatory cell recruitment. The anti-proliferative effect of alitretinoin is related to RARs, whereas retinoid X receptors (RXRs) mediate its apoptotic activity. Oral alitretinoin showed efficacy as measured by the Dermatology Life Quality Index (DLQI) and Sartorius scoring in four patients aged 30–75 years. Quality of Life (QoL) scores as calculated by the DLQI at week 0, 12, and 24 were 19.7, 8.0, and 6.7, respectively, achieving an average reduction of 13 points by week 24. Similar results were achieved using the Sartorius scale, with an average reduction of 15.7 [32].

Matusiak, et al. evaluated the efficacy of acitretin monotherapy in 17 patients with chronic and recalcitrant HS. The patients received acitretin for up to 9 months. Out of 17 patients, 53% completed 9 months of treatment. The mean dose was 0.56 mg/kg/day [35].

In another retrospective study, a beneficial effect with isotretinoin (*n* = 39) was observed in 14 (35.9%) patients. It seemed that those with a history of pilonidal cysts are more likely to respond to isotretinoin [36]. On the contrary, cases of severe acne treated with isotretinoin have been shown to present with lesions similar to HS [37]. Thus, strong evidence is not available to support the use of isotretinoin in patients with HS. Moreover, there are no clear patient profiles in whom isotretinoin would be helpful [38].

According to the European guidelines, acitretin can be started as the treatment in the early stages of HS. It may also be used in the chronic stage of the disease with sinus tracts and scarring. The recommended dose of acitretin is 0.25 to 0.88 mg/kg daily with a duration ranging from 3 months to one year. Isotretinoin is not recommended for use in the treatment of HS by European guidelines [18].

A Brazilian Society of Dermatology consensus document recommends acitretin for the treatment of moderate to severe HS [20]. According to a Swiss expert panel, acitretin is recommended for the treatment of follicular occlusion in moderate to severe cases. Similarly, in chronic moderate to severe cases, acitretin is recommended [9].

## 4. Biologics

Subcutaneous adalimumab, a tumor necrosis factor-α blocker is used in patients with moderate to severe HS. Studies have shown that adalimumab provides a clinical response in a higher proportion of patients as compared to a placebo at week 12 [39]. PIONEER I and II, the phase 3 multicenter trials, enrolled 307 and 326 patients, respectively. Clinical response rates at week 12 were significantly higher in patients receiving adalimumab as compared to placebo groups. The clinical response rates in adalimumab groups in the PIONEER I and II trials were 41.8% and 58.9%, respectively, as compared to 26.0% and 27.6% in placebo groups, respectively. Significantly higher improvements in lesions, pain, and disease severity were observed in the adalimumab group at week 12 in PIONEER II only [40]. In women with moderate to severe HS, 40 mg of adalimumab given per week has shown a reduction in disease severity and pain in a higher percentage of patients as compared to 40 mg of adalimumab administered every other week or a placebo [41]. Adalimumab also has potential to improve quality of life and patient satisfaction. Adalimumab is generally well tolerated in patients with moderate to severe HS [36]. A systematic review and meta-analysis reported the superiority of adalimumab (administered weekly) over a placebo in decreasing the symptoms [42].

The literature search suggests limited safety concerns with long-term treatment provided patients are screened for infections, tuberculosis, and malignancy before the initiation of treatment [43,44]. If antibiotics do not provide a satisfactory response, adalimumab may be considered for the treatment of HS [10]. In a real-life experience, the effectiveness of adalimumab has been studied in an observational retrospective study involving patients with moderate to severe HS. There was significant improvement in the symptoms and severity of HS as well as quality of life in these patients. The percentage of patients showing a clinical response increased from 10.5% at week 4 to 63.2% at week 24. According to the Brazilian Society of Dermatology consensus statement, adalimumab has level I evidence and grade A recommendation for use in HS [20].

A placebo-controlled trial involving 38 patients reported the efficacy and safety of infliximab in HS. Infliximab was used at the dose of 5 mg/kg at weeks 0, 2, and 6 and every 8 weeks thereafter. A total of 26% and 5% of patients achieved at least 50% improvement in the infliximab and placebo groups, respectively. In this study, infliximab monotherapy was well tolerated [45]. The level of evidence and grade of recommendation for infliximab is II/B [20]. Methotrexate can be added with infliximab to improve the efficacy [20].

Because of several limitations in the studies, including small sample size and non-validated scales for evaluation, it is difficult to provide possible conclusions for the use of etanercept in HS [20]. Others biologics including ustekinumab, anakinra, canakinumab, and secukinumab also need more evidence in HS [20]. In patients with moderate to severe disease, adalimumab 40 mg SC weekly is recommended. Infliximab is given at the dose of 5 mg/kg IV at weeks 0, 2, and 6, then every 2 months [18]. The Brazilian Society of Dermatology consensus document recommends adalimumab for the treatment of moderate to severe HS. Surgery is required for inactive/cicatricial lesions [20]. According to a Swiss expert panel, adalimumab is recommended for the treatment of inflammation in moderate to severe cases. For the management of chronic moderate to severe cases, adalimumab is recommended [9].

## 5. Treatment Approach

Overall, a holistic approach is essential for the management of HS. The medical management consists of two components: disease management and comorbidities’ management. Patients should be counselled and educated about the nature of the disease, which is essential for improving compliance to therapy [46]. Moderate to severe cases need an individualized approach based on the patient presentation and response to previously used treatments.

Most of the patients with HS need combination therapy; however, research from well-designed clinical trials is lacking. Research with combination therapy in HS should be encouraged.

## 6. Conclusions

Though a number of drugs are available in a dermatologist’s armamentarium, still the management of hidradenitis suppurativa remains unsatisfactory. Management is subjective and the choice of drug depends on the clinical severity, patient profile, risk–benefit ratio, and the judgement of the treating dermatologist. This review details a number of promising drugs available, but there is a need for large-scale randomized controlled trials to conclusively establish the efficacy of various oral immunosuppressants in the management of HS.

In general, in the majority of patients, immunosuppressants are not very effective, hence the requirement of combination therapy, but most research studies have evaluated these medications as monotherapy. Future studies are required to evaluate their efficacy and safety as a component of combination therapy. We hope this review serves as a guide for physicians to choose the appropriate therapies and improve their patients’ overall treatment outcomes.

## Figures and Tables

**Table 1 jcm-11-00670-t001:** Immunosuppressive drugs used for management of hidradenitis suppurativa [9,11,12].

Drug	Mechanism of Action	Dose	Adverse Effects	Pregnancy	Special Points
Azathioprine	It inhibits the synthesis of DNA in immune effector cells	0.6–0.75 mg/kg/day orally	Myelosuppression, hepatotoxicity, and gastric irritation	Pregnancy category D drug, it readily crosses the placenta and is excreted in breastmilk and colostrum, so best avoided during lactation	It should not be used in persons with deficiency of enzyme thiopurine methyltransferase.
Cyclosporine	Calcineurin inhibitor	3–5 mg/kg/day orally	Renal dysfunction, hypertension, gingival hyperplasia, hyperkalemia, hyperuricemia, nausea, abdominal discomfort, tremor, headache, arthralgia	Pregnancy category C drug. It is excreted in breast milk and may interfere in cellular metabolism of nursing infant	Contraindicated in children less than 18 years of age, but it has been used in 2–16 year age group for atopic dermatitis, macrophage activation syndrome (MAS), cutaneous T cell lymphoma, past history of malignancies. Renal function, blood pressure, and serum potassium should be monitored regularly.
Corticosteroids	Anti-inflammatory effect due to inhibition of prostaglandin, leukotriene, and cytokine synthesis.	Low dose is used orally (dose is subjective) Prednisolone dose 0.5–0.7 mg/kg/day	Gastritis, headache, weight gain, Cushing syndrome, cataract, dyslipidemia, hypertension, diabetes mellitus, depression, increased susceptibility to infections	Pregnancy category C	Patients need to be monitored for blood pressure, blood glucose, weight, waist circumference, blood electrolytes, and any infections. Ophthalmic examination should be done every 6–12 months.
Dapsone	Exact mechanism is unclear, but has shown sporadic benefit possibly due to its antineutrophilic action	25–150 mg/day orally	Nausea, headache, insomnia, leukopenia, agranulocytosis, hemolytic anaemia, peripheral neuritis, hepatitis, drug hypersensitivity syndrome.	Pregnancy category C	Should be avoided in individuals with glucose 6 phosphate dehydrogenase (G6PD) deficiency.
Methotrexate	It inhibits dihydrofolic acid reductase and interferes with DNA synthesis, repair, and replication of cell	10–15 mg/week	Myelosuppressions and hepatotoxicity	Pregnancy category X	Liver function and platelet count should be monitored regularly. Women of child bearing age on methotrexate should use proper contraception, and men on methotrexate should be counselled regarding possible reversible oligospermia.
Retinoids	Apoptosis of different cells in the body and down-regulates the enzyme telomerase reverse transcriptase. Thus has an anti-inflammatory and anti-proliferative effect.	Isotretinoin 0.5–1 mg/kg/day orally	Cheilitis, xerosis, headache, myalgia, pyogenic granuloma, hair fall, depression, pseudotumor cerebri	Category X	Women of child bearing age should use two forms of contraceptives while on treatment with retinoids

## Data Availability

Not applicable.

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
