# Peer review of "Efficacy and Toxicity of Classical Immunosuppressants, Retinoids and Biologics in Hidradenitis Suppurativa"

_jcm, 2022, doi:10.3390/jcm11030670_

Round 1
Reviewer 1 Report
The paper describes pros and cons of current drugs treatment in patients with hidradenitis suppurativa. The manuscript is well written and clear. A figure with the number and criteria of selection of literature papers would be useful.
Minor suggestions:
-line 34: Chronic and recurrent: this characteristics can sound misleading... relapsing?
-line 37: inframammary and intermammary...and mammary areas?
-line 39: any % of familiarity from the literature?
-line 43: lifestyle factors (such as ...)...
-line 61: there is no cure for HS: this sounds too strong for me...
-line 90: with relevant literature;
-table 1: mechanism of action of dapsone is missing... in case it is not known, please add "unknown" or something;
-table 1: cyclosporine is also used in children with MAS (macrophage activation syndrome)
-line 152: maybe better to include this final sentence near the prednisolone dosage (line 137);
-line 203: RARs?
-line 218: RXRs?
-line 219: even DLQI is not specify despite usually used.
Author Response
Reviewer 1
The paper describes pros and cons of current drugs treatment in patients with hidradenitis suppurativa. The manuscript is well written and clear. A figure with the number and criteria of selection of literature papers would be useful.
Author response- As this a narrative review, we have not prepared the PRISMA chart and do not have the number of articles as requested. Have added few more keywords.
Minor suggestions:-
-line 34: Chronic and recurrent: this characteristics can sound misleading... relapsing?
Have made the necessary correction.
-line 37: inframammary and intermammary...and mammary areas?
Have made the necessary correction.
-line 39: any % of familiarity from the literature?
Rephrased the sentence. Changed ref number 2.
-line 43: lifestyle factors (such as ...)...
Have made the necessary correction.
-line 61: there is no cure for HS: this sounds too strong for me...
Have modified the sentence
-line 90: with relevant literature;
Corrected
-table 1: mechanism of action of dapsone is missing... in case it is not known, please add "unknown" or something;
Have included the MoA of Dapsone in Table 1
-table 1: cyclosporine is also used in children with MAS (macrophage activation syndrome)
Added this part
-line 152: maybe better to include this final sentence near the prednisolone dosage (line 137);
Thank you. Done as suggested
-line 203: RARs?
Have expanded (Retinoic acid receptor)
-line 218: RXRs?
Have expanded (Retinoid X receptor)
-line 219: even DLQI is not specify despite usually used.
This is true, we have reported what instruments the authors used in that particular study

Reviewer 2 Report
Kind regards, I think the article is interesting and offers good information about the disease and its treatment. I think it would be nice to add certain details of the search in the results section such as number of articles located or articles included in the review. Although it is not a systematic review, this information may be important to correctly characterize the content. Thanks a lot.
Author Response
Reviewer 2
Kind regards, I think the article is interesting and offers good information about the disease and its treatment. I think it would be nice to add certain details of the search in the results section such as number of articles located or articles included in the review. Although it is not a systematic review, this information may be important to correctly characterize the content. Thanks a lot.
Author response- As this a narrative review, we have not prepared the PRISMA chart and do not have the number of articles as requested. Have added few more keywords.

Reviewer 3 Report
This review on immune suppressors and biologics in HS is an interesting attempt. However, it is lacking the number of studies for each treatment (precising RCT or not, retrospective or prospective), the primary objective of each study, the assessment score, patients’ severity (Hurley score), the follow-up length and total number of patients tested with each medication (excluding combination with another treatment or not), as well as % response to treatment.
Specific comments:
Line 41: “Several authors consider loss of function of GS…” would be better phrased as ïƒ GS mutations have been associated with HS in 5% cases.
Line 45: Instead of hypothesizing about secondary infection, it would be more factual to write that HS lesions have been associated with a commensal flora by several studies from 4 different teams (Guet-Revillet, Ring H, Schneider and Naik) and that bacterial biofilms have been identified in HS lesions, justifying surgery, which should at least be cited in the introduction, in treatment approach or in conclusion as a necessary component in combination treatment. The current concept of a host-microbiome disease for HS should also probably be cited.
Line 49: seronegative but also seropositive spondyloarthropathies have been both described in association with HS (several references are available in the literature)
P3: corticosteroids “dose is subjective “: what do the authors mean?
Line 136: “Oral corticosteroids are often… and lead to remission”. A reference for this assertion of remission with only steroids is needed, do authors mean improvement ? Or remission in combination treatments?
Line 138: “However, in some cases withdrawal of cortciosteroids (typo to correct) may lead to relapse.”
Line 219: what “average depression scores” refers to ?
Line 291 What does HD refer to?
L ine 302 be
Author Response
Reviewer 3
This review on immune suppressors and biologics in HS is an interesting attempt. However, it is lacking the number of studies for each treatment (precising RCT or not, retrospective or prospective), the primary objective of each study, the assessment score, patients’ severity (Hurley score), the follow-up length and total number of patients tested with each medication (excluding combination with another treatment or not), as well as % response to treatment.
Thank you for your suggestion. However, we have not maintained this data and segregated it as it is a narrative review presenting an overview of the topic. WE plan to include these comments in the future projects.
GS mutations have been associated with HS in 5% cases.àSeveral authors consider loss of function of GS…” would be better phrased as
Corrected
Line 45: Instead of hypothesizing about secondary infection, it would be more factual to write that HS lesions have been associated with a commensal flora by several studies from 4 different teams (Guet-Revillet, Ring H, Schneider and Naik) and that bacterial biofilms have been identified in HS lesions, justifying surgery, which should at least be cited in the introduction, in treatment approach or in conclusion as a necessary component in combination treatment. The current concept of a host-microbiome disease for HS should also probably be cited.
Corrections made as suggested.
Surgical option has been mentioned, line 69. However, both surgical and antibiotic therapy are beyond the scope of this review.
Line 49: seronegative but also seropositive spondyloarthropathies have been both described in association with HS (several references are available in the literature)
Have added serpositive…its mentioned in ref 2 (new ref replacing the older one).
P3: corticosteroids “dose is subjective “: what do the authors mean?
Have omitted this part. Thank you,
Line 136: “Oral corticosteroids are often… and lead to remission”. A reference for this assertion of remission with only steroids is needed, do authors mean improvement ? Or remission in combination treatments?
Have corrected as suggested
“However, in some cases withdrawal of cortciosteroids (typo to correct) may lead to relapse.”
Have corrected the typographic error
Line 219: what “average depression scores” refers to ?
Sorry for the error. Have replaced it to ‘Qoality of life (QoL) socres
Line 291 What does HD refer to?
It will be HS. Corrected
Line 302 be
Corrected
